# Dietary diversity and physical activity as risk factors of abdominal obesity among adults in Dilla town, Ethiopia

**Tinsae Shemelise Tesfaye**[1☯], **Tadesse Mekonen Zeleke**[2☯]*, **Wagaye Alemu**[1],
**Dirshaye Argaw**[1], **Tadesse Kebebe Bedane**[2]

**1** Public Health Department, College of Medicine and Health Science, Dilla University, Dilla, Ethiopia, **2** Food Science and Nutrition Directorate, Ethiopian Public Health Institute, Addis Ababa, Ethiopia

☯ These authors contributed equally to this work.
* anguach257@gmail.com

**Data Availability Statement:** All the relevant data are within the manuscript and its supporting information files.

## Abstract

### Background

Globally, the prevalence of obesity is on the rise and has nearly tripled since 1975. In Ethiopia, despite not having well-documented evidence, abdominal obesity has been increasing dramatically, particularly in urban settings. Therefore, this study is intended to determine the prevalence and risk factors of abdominal obesity among adults in Dilla town, Ethiopia.

### Methods

A community-based cross-sectional study was conducted between January and February 2018 in Dilla Town. A multi-stage sampling technique was employed to recruit 663 adults. The study was conducted in accordance with the World Health Organization (WHO) STEP wise approach. Waist circumference was measured using a flexible metric tape mid-way between the lowest rib and iliac crest with the participant standing at the end of gentle expiration. Abdominal obesity was determined using the International Diabetes Federation cut-off. A logistic regression model was fitted to identify risk factors of abdominal obesity. Adjusted odds ratio (AOR) with corresponding 95% confidence interval (CI) was calculated to show the strength of association.

### Results

A total of 634 adults participated in the study with a response rate of 95.6%. This study revealed that 155 (24.4%) [95% CI: (21.50, 27.80)] adults were abdominally obese. Higher odds of being abdominally obese were noted among adults with a high [AOR = 4.61, 95% CI: (2.51–8.45)] and middle [AOR = 3.22, 95% CI: (1.76–5.88)] wealth rank, consuming less diversified diet [AOR = 2.05, 95% CI: (1.31–3.19)], physical inactivity [AOR = 2.68, 95% CI: (1.70–4.22)] and being female [AOR = 1.92, 95% CI: (1.13–3.28)].

**Funding:** TST and WA received funding from Dilla University, which is provided only for staff. There is no grant number or URL website for this funding. TST, WA and DA received a salary from the funding institution, yet it is not for the purpose of conducting this study. The funder had no role in study design, data collection and analysis, decision to publish, or preparation of the manuscript.

**Competing interests:** The authors have declared that no competing interests exist in this work.

## Conclusions

The prevalence of abdominal obesity among adults in Dilla town is considerably high, and became an emerging nutrition related problem. Being in the middle and high wealth rank, physical inactivity, consuming less diversified diet, and being female were the risk factors of abdominal obesity.

## Introduction

Obesity is a preventable chronic disease affecting people across all ages, sexes, and ethnicities. Globally, the prevalence of obesity is on the rise and has nearly tripled between 1975 and 2016. In 2016, overweight affects more than 1.9 billion adults; of these, over 650 million were obese [1]. The epidemiological significance of obesity has also been growing in Ethiopia; between 1990 and 2009, obesity grew substantially from 0.7 to 2% among men and 6 to 10.8% among women [2,3].

Different risk factors contribute to the development of obesity; genetic, biological, individual, social, and environmental factors, which affect weight gain through the mediators of energy intake and expenditure. Obesity is associated with an increased risk of nearly every chronic condition, from diabetes, to dyslipidemia and poor mental health. Its impact on the risk of stroke and cardiovascular disease, certain cancers, and osteoarthritis are significant [4]. Furthermore, various studies have shown that the distribution of fat is more critical than the total amount of fat alone. Increased abdominal fat accumulation was found to be an independent risk factor for type 2 diabetes mellitus and cardiovascular risk conditions, such as coronary artery disease, stroke, and hypertension. It is also known that abdominal obesity is a more important risk factor for coronary heart disease than overall obesity. Visceral fat accumulation is associated with increased secretion of free fatty acids, hyperinsulinemia, insulin resistance, hypertension, and dyslipidemia [5–7].

As a result of rapid epidemiological and economic transition attributed to increased urbanization and globalization, many sub-Saharan African countries are experiencing lifestyle and behavioral changes such as unhealthy diet, physical inactivity, and increased tobacco use. These behavioral risk factors are responsible for substantial increases in the prevalence of intermediate cardiovascular disease risk factors including hypertension and obesity [8,9].

Despite not having well-documented evidence, abdominal obesity has been increasing dramatically, particularly among urban settings in Ethiopia. In the present study, we intended to assess the prevalence and risk factors of abdominal obesity among adults in Dilla town, Ethiopia.

## Materials and methods

### Study design, setting and period

This study was conducted among adults in Dilla town. The town is located in the southern part of Ethiopia. Dilla town is the capital of the Gedeo Administrative zone in South nations, nationalities, and peoples' region at a distance of 359 kilometers from Addis Ababa, the capital city of Ethiopia. The town is divided into nine kebeles (the smallest administrative unit in Ethiopia). These include Boiti, Harsu, Weldena, Buno, Hasse Della, Haroressa, Bareda, Haroke, and Odayaa kebeles. A community-based cross-sectional study was conducted among 663 adults aged 18 to 64 years between January and February 2018.

## Source and study population

Adults whose age was between 18 and 64 years in the selected kebeles were eligible for the study. We excluded adults with visible or self-reported pregnancy and body deformity around the abdomen.

## Sample size and sampling technique

The sample size was determined using a single population proportion formula. The formula estimates the minimum sample size required to determine the proportion in a source population. Specifications made during the computation were: 50% expected prevalence of abdominal obesity, 95% confidence level, 5% margin of error, 15% compensation for possible non-response, and design effect of 1.5. By using these specifications, the final sample size was 663 adults. Study participants were selected using a multistage sampling technique [10]. Initially, 4 kebeles (Harsu, Odaya, Buno, and Woldena) were randomly selected from the existing 9 kebeles in Dilla town, and the total sample size was distributed to the selected kebeles, proportional to their population size. Households were chosen by systematic random sampling approach. Bottle spinning method, in the middle of each selected kebeles was used to select the first household. A random number was identified within the sampling interval and households in the direction of the bottle head were counted until the selected number was reached then. The next household was selected by adding the sampling interval to the randomly selected number. When multiple eligible adults were available within the household, an adult aged 18–64 years were chosen randomly from the same household.

## Variables

Abdominal obesity was taken as the dependent variable.

## Socio-demographic and lifestyle variables

The questionnaire used in this study was adapted from the WHO-STEP wise questionnaire for chronic non-communicable disease [11], consists of socio-demographic information, dietary intakes, physical activity, and health risky behavior questions.

The food frequency questionnaire modified from WHO-STEP wise approach was used to assess the dietary habits of adults. Adults were asked to report their frequency of consumption for one usual week over the past 12 months. For dietary diversity, a simple count of food groups was used to calculate dietary diversity score (DDS). DDS was ranked into two groups, those who consumed six and above (high DDS) and less than six (low DDS). Those who drank alcohol in the past 12 months at least 3 days per week and above were taken as current alcohol users. Current smokers were defined as those who smoke at least 1 cigarette per day (at least 7 cigarettes per week) [11]. Similarly, current chat chewers were defined as those who had been chew chat for more than 6 months and chew chat 30 days preceding the study.

The survey included questions about the frequency of practicing physical activity during a typical week. The Global Physical Activity Questionnaire developed by WHO was used to assess physical activity patterns among adults. The activity level of adults was evaluated according to the standard WHO total physical activity calculation guide and the level of total physical activity was categorized as physically active and physically inactive. Adults were considered to be physically active if their total physical activity MET (Metabolic Equivalent) minute/week had greater than and equal to 600 MET-minutes, whereas physically in-active if the total physical activity MET minute/week was less than 600 MET-minutes [11].

Wealth index was generated using principal component analysis. The scores of 18 selected groups of assets and utilities were translated into latent factors. The first factor that explained most of the variation was used to group study subjects into three ranked wealth groups (low, middle, and high).

### Data collection and measurement

All the adults were interviewed for their socio-demographic information, dietary intakes, physical activity, and risky health behavior. Anthropometric measurement was taken at the end of the interview. Data were collected by six diploma nurses and two supervisors.

The data quality was ensured during tool development, data collection, coding, entry, and analysis. The questionnaire was initially prepared in English and translated into Amharic and Gedeoffa and re-translated into English. The questionnaire was pretested among randomly selected adults that were not included in the main survey. Its validity was examined among 60 adults, which showed that it was acceptable and understandable. Additionally, training was given to the data collectors and supervisors on the questionnaire, methods, and procedures of taking measurements on waist circumference. Supervisors made spot checking and review of all completed questionnaires to ensure completeness and consistency of the information collected. Supervisors also re-took measurements on 10% of the study participants from each data collector to check the reliability of the measurements.

WC was measured in a standing position midway between the lower rib margin and the anterior superior iliac crest in the horizontal plane using a flexible plastic metric tape to the nearest 0.1 cm. The measurement was taken when the participant was at the end of the gentle expiration, after taking a deep inhalation with the tape snug but ensuring that it was not compressing the skin. WC was measured in duplicate and the average value was used for analyses. Abdominal obesity was defined as waist circumstance for European region recommendation as per International Diabetes Federation, greater than and equal to 94 cm for males and greater than and equals to 80 cm for females [12].

### Data analysis

The data were analyzed using SPSS for Windows, version 20.0. Descriptive statistics were conducted and results were presented using tables and figures. Logistic regression was performed to assess the association between the factors of interest and abdominal obesity. Binary logistic regression was conducted to select candidate variables (P-value < 0.25) for multiple logistic regression. In the multiple logistic regression, variables having $P$-value < 0.05 were declared as statistically significant variables. AOR and 95% CI were calculated to measure the strength of associations.

### Ethical statement

Ethical clearance was obtained from the Institutional Review Board of Dilla University. The study was also done following the declaration of Helsinki. The necessary permission was also obtained from the Gedeo Zone health and administrative offices, and from the selected kebele offices. Informed verbal consent was obtained before the start of data collection. The participants were also assured that they had the full right to participate or withdraw at any time during the study.

## Results

### Socio-demographic characteristics

Among the study participants, nearly half (45.0%) of them were in the age group of 25–34. More than half (56.0%) of the participants were illiterate and had not gone further than

**Table 1. Socio-demographic characteristics of study participants, Dilla town, Ethiopia, 2018.**

| Characteristics | | Number (n = 634) | Percent |
|---|---|---|---|
| **Sex** | Female | 362 | 57.1 |
| | Male | 272 | 42.9 |
| **Age** | 18–24 | 130 | 20.5 |
| | 25–34 | 285 | 45.0 |
| | 35–44 | 112 | 17.7 |
| | 45–54 | 80 | 12.6 |
| | 55–64 | 27 | 4.3 |
| **Ethnicity** | Gedio | 281 | 44.3 |
| | Gurage | 159 | 25.1 |
| | Amhara | 91 | 14.4 |
| | Oromo | 71 | 11.2 |
| | Others | 32 | 5.0 |
| **Educational status** | No formal school | 119 | 18.8 |
| | Primary | 236 | 37.2 |
| | Secondary | 196 | 30.9 |
| | College and above | 83 | 13.1 |
| **Marital status** | Married | 323 | 50.9 |
| | Single | 247 | 39.0 |
| | Divorced | 34 | 5.4 |
| | Widowed | 30 | 4.7 |
| **Religion** | Protestant | 260 | 41.0 |
| | Orthodox | 249 | 39.3 |
| | Muslim | 96 | 15.1 |
| | Others | 29 | 4.6 |
| **Occupation** | Gov't employee | 244 | 38.5 |
| | Non-Gov't employee | 231 | 36.4 |
| | Unemployed | 98 | 15.5 |
| | Student | 61 | 9.6 |
| **Wealth Index** | Low | 217 | 34.2 |
| | Middle | 218 | 34.4 |
| | High | 199 | 31.4 |

primary education. Nearly 3 out of 4 (74.9%) adults were either government or private employees and 50.9% lived in marriage. Regarding their wealth status, about 65.8% of them were in the middle and above wealth rank (Table 1).

## Dietary characteristics

Table 2 describes the participants' dietary characteristics. More than half (56.5%) of participants consumed less diversified food, while more men (60.7) consumed less diversified food than women (53.3). About half of the women (55.2) and men (49.6) didn't skip breakfast, while more than half (58.7) of the participants had no snack.

## Behavioral characteristics

Of the total adults, one hundred fifty-five (24.4%) [95% CI: (21.50, 27.80)] had abdominal obesity. The prevalence of abdominal obesity was recorded in 99 (27.3%) females, which was higher compared to males, 56 (20.6%). Concerning behavioral risk factors by gender, nearly three quarters

**Table 2. Dietary characteristics of study participants, Dilla town, Ethiopia, 2018.**

| Dietary related characteristics | | Female number (%) | Male number (%) |
|---|---|---|---|
| **DDS** | High | 169 (46.7) | 107 (39.3) |
| | Low | 193 (53.3) | 165 (60.7) |
| **Breakfast Skipping** | Didn't skip | 200 (55.2) | 135 (49.6) |
| | Once | 35 (9.7) | 38 (14.0) |
| | Twice | 67 (18.5) | 41 (15.1) |
| | Three and more times | 60 (16.6) | 58 (21.3) |
| **Snack** | Yes | 153 (42.3) | 109 (40.1) |
| | No | 209 (57.7) | 163 (59.9) |

(73.2%) of females never chewed chat. About 14.2% and 21.1% of adults accompanied chat chewing with cigarette smoking and alcohol consumption, respectively. Only 2.8% of the women were smokers, while more than a third of the men either smoked during the survey or had smoked before. In both sexes, more than two-thirds of the participants were physically active (Table 3).

## Risk factors of abdominal obesity among adults

The final multivariable logistic regression model analysis showed that being in middle and high wealth rank, consuming lower dietary diversity, physical inactivity and being female were the risk factors for abdominal obesity. Adults who consumed a less diversified diet were twice as likely to get abdominal obesity as those who were eating a more diversified diet [AOR = 2.05, 95% CI: (1.31–3.19)]. The study also showed that improvement in the distribution of wealth index increased the chance of abdominal obesity. Adults with high and middle categories of wealth rank were 4.6 [AOR = 4.61, 95% CI: (2.51–8.45)] and 3.2 [AOR = 3.22, 95% CI: (1.76–5.88)] times more likely to get abdominal obesity than those with low wealth rank, respectively. Besides, physically inactive adults were 2.7 [AOR = 2.68, 95% CI: (1.70–4.22)] times more likely to have abdominal obesity than physically active ones. Furthermore, women were almost twice [AOR = 1.92, 95% CI: (1.13–3.28)] abdominally obese as compared to men's (Table 4).

## Discussion

Abdominal obesity is one of the emerging nutritional problems in developing countries, including Ethiopia. It is related to the risk of cardio-metabolic comorbidities and taking measures to control abdominal obesity will help to reduce threats of the problem.

**Table 3. Distribution of abdominal obesity and behavioral risk factors by gender among participants, Dilla town, Ethiopia, 2018.**

| Behavioral risk-related characteristics | | Female number (%) | Male number (%) |
|---|---|---|---|
| **Abdominal obesity** | Yes | 99 (27.3) | 56 (20.6) |
| | No | 263 (72.7) | 216 (79.4) |
| **Chat chewing** | Current and Ever | 97 (26.8) | 197 (72.4) |
| | Never | 265 (73.2) | 75 (27.6) |
| **Alcohol drinking** | Yes | 113 (31.2) | 121 (44.5) |
| | No | 249 (68.8) | 151 (55.5) |
| **Smoking** | Current and Ever | 10 (2.8) | 100 (36.8) |
| | Never | 352 (97.2) | 172 (63.2) |
| **Physical activity** | Active | 251 (69.3) | 202 (74.3) |
| | Inactive | 111 (30.7) | 70 (25.7) |

**Table 4. Risk factors of abdominal obesity among study participants, Dilla town, Ethiopia, 2018.**

| | Abdominal obesity | | | |
|---|---|---|---|---|
| Variables | Yes | No | AOR (95%) | P-Value |
| **Sex** | | | | |
| Female | 99 | 263 | 1.92 (1.13–3.28) | 0.02** |
| Male | 56 | 216 | 1 | |
| **Dietary diversity** | | | | |
| Low | 106 | 252 | 2.05 (1.31–3.19) | 0.00** |
| High | 49 | 227 | 1 | |
| **Wealth index** | | | | |
| Low | 21 | 196 | 1 | |
| Middle | 59 | 159 | 3.22 (1.76–5.88) | 0.00** |
| High | 75 | 124 | 4.61 (2.51–8.45) | 0.00** |
| **Physical activity** | | | | |
| Inactive | 71 | 110 | 2.68 (1.70–4.22) | 0.00** |
| Active | 84 | 369 | 1 | |

**Significantly associated variables at p-value < 0.05, 1—Reference group.

Abbreviations: AOR—Adjusted odds ratio.

The current study revealed that one-fourth of adults 24.4% [95% CI: (21.50, 27.80)] had abdominal obesity. It has been demonstrated in a previous study conducted in Addis Ababa, Ethiopia, that the prevalence of abdominal obesity was 19.6% [13], which is lower as compared to the current study. Similar lower findings were reported in Benin (15.5%) and Uganda (11.8%) [14,15]. However, the prevalence of abdominal obesity was much higher in studies conducted in various countries [16–20] as compared to this study. Similar to the previous studies [13–17,19,20], this study also found a higher prevalence of abdominal obesity in females (27.3%) than in males (20.6%).

Although genetics play an important role in the obesity epidemic, other factors play an important role too, including nutritional transition as a result of urbanization and westerniza-tion, which promotes the neglect of traditional healthy diets, less physical activity and the consumption of westernized, energy-dense foods and sugar-sweetened beverages [16].

In the present study, middle and higher wealth rank, eating a less diversified diet, physical inactivity and being female were independent risk factors for abdominal obesity. Similarly, women had higher measures of abdominal obesity compared to men. Our findings confirm what has been reported in other studies [17,19–21]. Likely reasons for this observation include that women are physiologically more predisposed to overweight and obesity due to the changes occurring during the reproductive years, and women have been reported to engage in less physically strenuous/demanding activities in the moderate to vigorous range compared to men in the same setting [21].

Most importantly, as the economic status improved, the prevalence of abdominal obesity also increased, and it was found to be statistically significant (P<0.00). The prevalence of abdominal obesity was significantly higher in the second and third quartiles of household wealth rank as compared to the first quartile. This finding was consistent with the studies done in India and Cameroon [22–24]. Economic inequalities in health have been attributed to several different mechanisms, including change in lifestyle (unhealthy behaviors, consumption of energy-dense foods, and sedentary way of life) and inadequate access to health care.

Regarding the relationship between physical activity and abdominal obesity, the result showed an indirect association. Physically inactive adults were more likely to have abdominal obesity than physically active ones. This finding is supported by plenty of literature. For instance, studies done in Ghana, adults with lower levels of physical activity demonstrated a higher likelihood of abdominal obesity [25] and Nigeria [26]. Moreover, another study done in Poland revealed that the risk of abdominal obesity was significantly lower among adolescents who declared higher physical activity [27]. On the contrary, a study conducted among Spanish adults reported that more time spent in vigorous physical activity, but not in moderate-vigorous physical activity, was associated with a lower risk of abdominal obesity [28]. Physical activity is a key determinant of energy expenditure. Therefore, based on our findings, it is highly advisable to be physically active, which includes light to high intensity, to prevent abdominal obesity.

Consumption of a diversified diet is another factor that has been explored in this study. Interestingly, adults who consumed less diversified diets were more likely to be abdominally obese than those who were eating a more diversified diet. A finding from the NHANES study in the United States supports this finding, dietary diversity score has an inverse association with indicators of body adiposity in both sexes and indicated that healthy food varieties can protect against excess adiposity [29]. Furthermore, a cross-sectional study among Iranian women aged 18 to 28 years old found that a higher dietary diversity quartile was associated with lower odds of both general and abdominal obesity [30] and another study among Iranian adults with pre-diabetes revealed that DDS was inversely associated with metabolic syndrome [31]. Moreover, a study among urban South Indians showed that increased intake of fruits and vegetables could play a protective role against obesity-associated metabolic risk factors [32]. However, community-based cross-sectional studies conducted among rural Asian Indians [18] and Sri Lankan [33] showed a positive association between abdominal obesity and DDS. In both studies, abdominally obese participants had a higher DDS score compared to non-abdominally obese groups. These different results could be due to different methods and populations used in assessing abdominal obesity, dietary intake, and determination of DDS [34].

Even though significant association was noted between abdominal obesity and risk factor like age [35]. Our study revealed no significant association between abdominal obesity and age, marital status, smoking status, alcohol consumption, or chat chewing.

## Strength and limitation of the study

This study has several strengths including the use of calibrated instrument, standardization during training, and close supervision and spot checking during data collection. Our study also has some limitations that need to be considered. First, as the scope of this study is limited to behavioral and physical measurements, and does not include biochemical measurements. Second, there could be differences in abdominal obesity by season in which the current study was not able to assess. Third, the cross-sectional design of our study only allowed the assessment of the associations between abdominal obesity and risk factors rather than causal links.

## Conclusions

In conclusion, the findings revealed a high prevalence of abdominal obesity in the study area. Being in the middle and high wealth rank, physical inactivity, consuming less diversified diet, and being female were the risk factors of abdominal obesity. The findings highlight that there is an urgent need for evidence-based prevention and management of abdominal obesity and its associated factors among adults in Dilla town. Consequently, all concerned stakeholders

need to strengthen existing strategies related to the delivery of nutrition services for non-communicable diseases with especial attention to women.

## Supporting information

**S1 Appendix. Wealth index assets and utilities, Dilla town, Ethiopia, 2018.**
(PDF)

## Acknowledgments

We would like to express our deepest gratitude to Dilla University, participants, and field staff.

## Author Contributions

**Conceptualization:** Tinsae Shemelise Tesfaye, Tadesse Mekonen Zeleke.

**Data curation:** Tinsae Shemelise Tesfaye, Tadesse Mekonen Zeleke.

**Formal analysis:** Tinsae Shemelise Tesfaye, Tadesse Mekonen Zeleke, Dirshaye Argaw, Tadesse Kebebe Bedane.

**Funding acquisition:** Tinsae Shemelise Tesfaye, Wagaye Alemu.

**Investigation:** Tinsae Shemelise Tesfaye, Tadesse Mekonen Zeleke, Wagaye Alemu.

**Methodology:** Tinsae Shemelise Tesfaye, Tadesse Mekonen Zeleke.

**Project administration:** Tinsae Shemelise Tesfaye, Tadesse Mekonen Zeleke.

**Resources:** Tinsae Shemelise Tesfaye, Tadesse Mekonen Zeleke.

**Software:** Tinsae Shemelise Tesfaye, Tadesse Mekonen Zeleke, Wagaye Alemu.

**Supervision:** Tinsae Shemelise Tesfaye, Tadesse Mekonen Zeleke, Wagaye Alemu, Dirshaye Argaw.

**Validation:** Dirshaye Argaw, Tadesse Kebebe Bedane.

**Visualization:** Tinsae Shemelise Tesfaye, Tadesse Mekonen Zeleke.

**Writing – original draft:** Tinsae Shemelise Tesfaye, Tadesse Mekonen Zeleke, Wagaye Alemu, Tadesse Kebebe Bedane.

**Writing – review & editing:** Tinsae Shemelise Tesfaye, Tadesse Mekonen Zeleke, Dirshaye Argaw, Tadesse Kebebe Bedane.

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
