## [Decision Letter · Decision Letter 0]

19 May 2020

PONE-D-20-08295

Dietary diversity and physical activity as predictors of abdominal obesity among adults in Dilla town, Ethiopia

PLOS ONE

Dear Mr. Zeleke,

Thank you for submitting your manuscript to PLOS ONE. After careful consideration, we feel that it has merit but does not fully meet PLOS ONE’s publication criteria as it currently stands. Therefore, we invite you to submit a revised version of the manuscript that addresses the points raised during the review process.

We would appreciate receiving your revised manuscript by Jul 03 2020 11:59PM. To enhance the reproducibility of your results, we recommend that if applicable you deposit your laboratory protocols in protocols.io, where a protocol can be assigned its own identifier (DOI) such that it can be cited independently in the future. For instructions see: http://journals.plos.org/plosone/s/submission-guidelines#loc-laboratory-protocols

We look forward to receiving your revised manuscript.

Kind regards,

Sabine Rohrmann

Academic Editor

PLOS ONE

Journal Requirements:

Additional Editor Comments (if provided):

Reviewers' comments:

Reviewer's Responses to Questions

**Comments to the Author**

1. Is the manuscript technically sound, and do the data support the conclusions?

Reviewer #1: Yes

Reviewer #2: Yes

2. Has the statistical analysis been performed appropriately and rigorously? 

Reviewer #1: No

Reviewer #2: Yes

3. Have the authors made all data underlying the findings in their manuscript fully available?

Reviewer #1: Yes

Reviewer #2: Yes

4. Is the manuscript presented in an intelligible fashion and written in standard English?

Reviewer #1: Yes

Reviewer #2: Yes

5. Review Comments to the Author

Reviewer #1: Manuscript PONE-D-20-08295: Dietary diversity and physical activity as predictors of abdominal obesity among adults in Dilla town, Ethiopia

It is interesting that the authors assessed obesity in such an area where obesity is often not considered a problem. However, obesity is a global issue today. Appreciating the good efforts made, I have some concerns about the study that are outlined below.

- Throughout the document, it would be good to replace predictors with risk factors, associated factors, influencing factor, etc… Predictors sounds that the authors are developing a prediction model, not performing a regression model.

Line 15- Background/abstract: change “…mortality” to morbidity, obesity doesn’t cause mortality directly

- What was the obesity prevalence by sex, which I think is important to reflect? Obesity is not the same by sex, and this is all known. It would thus be good to present all results stratified by sex.

Lines 30-31: The results of the regression are strange. You reported that the wealth index was positively associated with an increase in obesity, with the OR of 4.75 (>1), which is okay. This is consistent with other studies in developing countries although the association may be the opposite in some other countries, especially in Europe. In the same model, I expected that the protective effect of food diversity and physical activity would be <1 but it is now >1. I got it where it went wrong. For example, you interpreted "physically active is protective with an OR of 2.53" This is not correct. What you have tested in your model is "being inactive", instead of “being active”; that is why you got OR of 2.53 which is >1. The interpretation is wrong as well and also that you would have to reverse the code of "physically active and inactive" and "high food variety and low food variety" in your model. Then you should find <1 OR, where (1-OR) would be the percentage that could be averted due to activity and high food variety.

- Line 80: “…whose ages was”

- Your sample should have been estimated on the basis of the assumption that the prevalence of obesity is different in men and women, rather than with a single proportion assumption–in other words, with two prevalence assumptions. This now seems irreversible. But well, it still seems the size you have already is sufficient to estimate the sample size for men and women separately. This is because you have used a prevalence of 0.5, which is much higher than the actual prevalence leading a larger sample size. If you did it correctly, i.e. taking two prevalence points for men and women, you would get a similar sample size (i.e. with a prevalence of, say 0.25, not just 0.5. I don’t see any stratified results by sex, especially for prevalence. Please do so.

- Lines 151-152: What is the point of performing a logistic regression for each risk factor one by one? It makes no sense, except if you are in a critical situation where the sample size is small given that you have many variables to enter into your model. In fact, it is not recommended that you run this regression for each risk factor one by one. In some cases, a variable may not be significantly associated with an outcome if you fit it in the model alone, but it could be an effect modifier (and its effect could be significant and clinically important) when you fit the variable in your model with other covariates. If you have excluded some variables in that step, please include them in your model and see the results.

- Line 180: “chewed”

- Line 181: accompanied”

- Line 182: Only 2.8% of the women were smokers, while more than a third of the men either smoked during the survey or had smokers before.

- Line 183-84: more than two-third of the participants were physically active

- Have you checked if your covariates were correlated? They seem to be quite a few of them.

Reviewer #2: Thank you for the manuscript. The authors conducted a cross-sectional study in southern Ethiopia, investigating prevalence and determinants of abdominal obesity. They observed a roughly 25% prevalence of obesity; wealth, dietary diversity and physical activity were its determinants. I have some comments and suggest major revision before acceptance.

Abstract:

I would suggest to add “waist circumference” in the methods. The authors mention abdominal obesity but no parameter.

I would expect to read AOR < 1 if the authors write about protectivity (higher dietary diversity and being physically active), however the authors state that “having higher dietary diversity (AOR = 1.89…) and performing physical activity (AOR=2.53…) were found to be protective against abdominal obesity.= I would suggest to rewrite the sentence or invert the AOR.

Introduction

Reference 1 does not seem to be appropriate here (the reference mainly reflects China, but the authors first sentence talks globally). A WHO reference might be more adequate here.

Line 46: Cancer is also one of the major possible consequences of obesity, please do add that to the list of metabolic complications associated with obesity (although cancer is not a metabolic complication but a complex disease, and obesity one of the major risk factors globally speaking).

Line 50: more critical for what? Please detail.

Materials and Methods

Line 89: Would the authors have a reference available for their choice of sampling technique? It seems to be the most appropriate in this setting but a reference would strengthen the authors choice and handling. And how was the one adult from a household chosen, if several adults belonged to a household and were generally eligible? I fear that specific members were chosen (either overweight, underweight or – to be on the safe side – normal weight) – and if this occurred several times, I would question the validity of the current study (and also the 25% prevalence of obesity in the population).

Line 122: What were the 18 selected assets and utilities to calculate the wealth index? This might be a possible supplementary table.

Line 130: I appreciate the tremendous efforts undertaken by the authors/personal to strengthen data and study quality. This was very well done.

Results:

In general, I would suggest to write numbers with two digits, the authors are mixing the number of digits (i.e. OR and AOR with two, 95%CI with three, p-values even more).

Line 168: Wealth Index: As depicted in Table 1, distribution across the three wealth indices was roughly equal. However, the authors write that two thirds were in the middle or above wealth rank – the other view would also be correct, i.e. two thirds were in the middle or below wealth rank. Both are subjective and I would refrain from applying subjective descriptions in the results section.

Table 3: Why not add prevalence and % of obesity into this table?

Line 191: As depicted in Table 4, sex and martial, smoking and chat chewing status never were associated with abdominal obesity, please correct.

Discussion:

The discussion would benefit from a native English reader correcting the grammar.

Thank you for comparing your baseline characteristics to other studies conducted in Ethiopia or further comparable countries.

Line 263: Please add your strengths here, such as your study and data quality protocol. And a limitation might be the cross-sectional design of the study, i.e. causality cannot be investigated. Which adds to further research that is needed.

6. PLOS authors have the option to publish the peer review history of their article (what does this mean?). If published, this will include your full peer review and any attached files.

Reviewer #1: No

Reviewer #2: No

---

## [Author Response · Author response to Decision Letter 0]

11 Jun 2020

Reviewer #1 

Throughout the document, it would be good to replace predictors with risk factors, associated factors, influencing factor, etc… predictors sounds that the authors are developing a prediction model, not performing a regression model.

Well accepted and addressed in the whole parts of the document including the title.

Line 15- Background/abstract: change “…mortality” to morbidity, obesity doesn’t cause mortality directly

Well said, obesity is linked to mortality rather than directly causing death. Yet the whole paragraph is modified with the latest WHO reference.

What was the obesity prevalence by sex, which I think is important to reflect? Obesity is not the same by sex, and this is all known. It would thus be good to present all results stratified by sex.

Agreed and prevalence of abdominal obesity is stratified by sex (Table 3). Variables under dietary-related characteristics are stratified as well (Table 2).

Lines 30-31: The results of the regression are strange. You reported that the wealth index was positively associated with an increase in obesity, with the OR of 4.75 (>1), which is okay. This is consistent with other studies in developing countries although the association may be the opposite in some other countries, especially in Europe. In the same model, I expected that the protective effect of food diversity and physical activity would be <1 but it is now >1. I got it where it went wrong. For example, you interpreted "physically active is protective with an OR of 2.53" This is not correct. What you have tested in your model is "being inactive", instead of “being active”; that is why you got OR of 2.53 which is >1. The interpretation is wrong as well and also that you would have to reverse the code of "physically active and inactive" and "high food variety and low food variety" in your model. Then you should find <1 OR, where (1-OR) would be the percentage that could be averted due to activity and high food variety.

We agree that interpretation of our result appears contrary to the finding and we have now rewritten with the following statement in the whole parts of the document (abstract, results and discussion), “The higher odds of being abdominally obese were noted among adults with a higher [AOR = 4.75, 95% CI: (2.62-8.60)] and middle [AOR = 3.12, 95% CI: (1.74-5.59)] categories of wealth rank, having lower dietary diversity [AOR = 1.89, 95% CI: (1.22-2.90)] and physical inactivity [AOR = 2.53, 95% CI: (1.64-3.90)].”

Line 80: “…whose ages was”

Has been changed accordingly.

Your sample should have been estimated on the basis of the assumption that the prevalence of obesity is different in men and women, rather than with a single proportion assumption–in other words, with two prevalence assumptions. This now seems irreversible. But well, it still seems the size you have already is sufficient to estimate the sample size for men and women separately. This is because you have used a prevalence of 0.5, which is much higher than the actual prevalence leading a larger sample size. If you did it correctly, i.e. taking two prevalence points for men and women, you would get a similar sample size (i.e. with a prevalence of, say 0.25, not just 0.5. I don’t see any stratified results by sex, especially for prevalence. Please do so.

Thank you for the information regarding sample size determination and the issue of stratifying the prevalence by sex is already addressed together with the comment given under result.

Lines 151-152: What is the point of performing a logistic regression for each risk factor one by one? It makes no sense, except if you are in a critical situation where the sample size is small given that you have many variables to enter into your model. In fact, it is not recommended that you run this regression for each risk factor one by one. In some cases, a variable may not be significantly associated with an outcome if you fit it in the model alone, but it could be an effect modifier (and its effect could be significant and clinically important) when you fit the variable in your model with other covariates. If you have excluded some variables in that step, please include them in your model and see the results.

Thank you very much for pointing it out and we really appreciate it. As per your valuable suggestion, we have tried to include those variables which were excluded initially based on their p-value (i.e <0.25 during bivariate). Consequently, sex (being women) became one of the variables in final model. And all the necessary modification was undertaken throughout the document. 

Line 180: “chewed”

Line 181: accompanied”

Line 182: Only 2.8% of the women were smokers, while more than a third of the men either smoked during the survey or had smokers before.

Line 183-84: more than two-third of the participants were physically active

Thank you for your suggestion and corrections are made.

Have you checked if your covariates were correlated? They seem to be quite a few of them.

Yes, we have checked it and they didn’t. Even we have tried to check for multicollinearity for each independent variable and their values range between 1 and 1.5.

Reviewer #2 

Abstract:

I would suggest to add “waist circumference” in the methods. The authors mention abdominal obesity but no parameter. 

We appreciate your suggestion. We have added the following sentence to the method part of the abstract. “waist circumference was measured using a flexible metric tape mid-way between the lowest rib and iliac crest with the participant standing at the end of gentle expiration.”

I would expect to read AOR < 1 if the authors write about protectivity (higher dietary diversity and being physically active), however the authors state that “having higher dietary diversity (AOR = 1.89…) and performing physical activity (AOR=2.53…) were found to be protective against abdominal obesity.= I would suggest to rewrite the sentence or invert the AOR. 

Well taken and it has been rewritten in the whole parts of the document (abstract, results and discussion) as “The higher odds of being abdominally obese were noted among adults with a high [AOR = 4.75, 95% CI: (2.62-8.60)] and middle [AOR = 3.12, 95% CI: (1.74-5.59)] categories of wealth rank, having lower dietary diversity [AOR = 1.89, 95% CI: (1.22-2.90)] and physical inactivity [AOR = 2.53, 95% CI: (1.64-3.90)].” 

Introduction

Reference 1 does not seem to be appropriate here (the reference mainly reflects China, but the authors first sentence talks globally). A WHO reference might be more adequate here.

Accepted and corrected accordingly. The paragraph is rewritten and an up-to-date WHO reference is used. 

Line 46: Cancer is also one of the major possible consequences of obesity, please do add that to the list of metabolic complications associated with obesity (although cancer is not a metabolic complication but a complex disease, and obesity one of the major risk factors globally speaking). Line 50: more critical for what? Please detail. Further explanation is added to detail about distribution of fat. 

We have edited and rewritten the whole paragraph as “different risk factors contribute to the development of obesity; genetic, biological, individual, social, and environmental factors, which affect weight gain through the mediators of energy intake and expenditure. Obesity is associated with an increased risk of nearly every chronic condition, from diabetes, to dyslipidemia, to poor mental health. Its impacts on the risk of stroke and cardiovascular disease, certain cancers, and osteoarthritis are significant. Furthermore, various studies have shown that the distribution of fat is more critical than the total amount of fat alone. Increased abdominal fat accumulation found to be an independent risk factor for type 2 diabetes mellitus and cardiovascular risk conditions, such as coronary artery disease, stroke, and hypertension. It is also known that abdominal obesity is a more important risk factor for coronary heart disease than overall obesity. Visceral fat accumulation is associated with increased secretion of free fatty acids, hyperinsulinemia, insulin resistance, hypertension, and dyslipidemia.”

Materials and Methods

Line 89: Would the authors have a reference available for their choice of sampling technique? It seems to be the most appropriate in this setting but a reference would strengthen the authors choice and handling. 

We also agree that multi stage sampling technique appears as the most appropriate technique. Yet, we have added a new reference. 

And how was the one adult from a household chosen, if several adults belonged to a household and were generally eligible? I fear that specific members were chosen (either overweight, underweight or – to be on the safe side – normal weight) – and if this occurred several times, I would question the validity of the current study (and also the 25% prevalence of obesity in the population). 

In the case of several eligible adults within a household, one adult was selected using a simple random method irrespective of his nutritional status. 

Line 122: What were the 18 selected assets and utilities to calculate the wealth index? This might be a possible supplementary table. 

We have now uploaded as supplementary information. 

Line 130: I appreciate the tremendous efforts undertaken by the authors/personal to strengthen data and study quality. This was very well done.

Thank you and we appreciate your comments. 

Results:

In general, I would suggest to write numbers with two digits, the authors are mixing the number of digits (i.e. OR and AOR with two, 95%CI with three, p-values even more).

Accepted and all the number of digits has changed to two digits (i.e. COR, AOR, 95% CI and P-values). 

Line 168: Wealth Index: As depicted in Table 1, distribution across the three wealth indices was roughly equal. However, the authors write that two thirds were in the middle or above wealth rank – the other view would also be correct, i.e. two thirds were in the middle or below wealth rank. Both are subjective and I would refrain from applying subjective descriptions in the results section.

 It is an important point and the statement has modified as “two-thirds were in the middle and above wealth rank” since middle and high wealth rank became two-third when they came together (added). 

Table 3: Why not add prevalence and % of obesity into this table?

Well accepted and the prevalence of obesity is stratified by sex and included in table 3.

Line 191: As depicted in Table 4, sex and martial, smoking and chat chewing status never were associated with abdominal obesity, please correct.

We strongly agree that these variables never were associated with abdominal obesity in the final model and corrected accordingly. 

Discussion:

The discussion would benefit from a native English reader correcting the grammar.

We appreciate that the reviewer asked for more accurate grammar and efforts were made to polish the discussion. 

Thank you for comparing your baseline characteristics to other studies conducted in Ethiopia or further comparable countries.

Line 263: Please add your strengths here, such as your study and data quality protocol. And a limitation might be the cross-sectional design of the study, i.e. causality cannot be investigated. Which adds to further research that is needed.

Great point. We have modified the strength and limitation paragraph.

---

## [Decision Letter · Decision Letter 1]

13 Jul 2020

Dietary diversity and physical activity as risk factors of abdominal obesity among adults in Dilla town, Ethiopia

PONE-D-20-08295R1

Dear Dr. Zeleke,

We’re pleased to inform you that your manuscript has been judged scientifically suitable for publication and will be formally accepted for publication once it meets all outstanding technical requirements.

Kind regards,

Sabine Rohrmann

Academic Editor

PLOS ONE

Additional Editor Comments (optional):

Reviewers' comments:

Reviewer's Responses to Questions

**Comments to the Author**

1. If the authors have adequately addressed your comments raised in a previous round of review and you feel that this manuscript is now acceptable for publication, you may indicate that here to bypass the “Comments to the Author” section, enter your conflict of interest statement in the “Confidential to Editor” section, and submit your "Accept" recommendation.

Reviewer #1: All comments have been addressed

Reviewer #2: All comments have been addressed

2. Is the manuscript technically sound, and do the data support the conclusions?

Reviewer #1: Yes

Reviewer #2: Yes

3. Has the statistical analysis been performed appropriately and rigorously? 

Reviewer #1: Yes

Reviewer #2: Yes

4. Have the authors made all data underlying the findings in their manuscript fully available?

Reviewer #1: Yes

Reviewer #2: Yes

5. Is the manuscript presented in an intelligible fashion and written in standard English?

Reviewer #1: Yes

Reviewer #2: Yes

6. Review Comments to the Author

Reviewer #1: (No Response)

Reviewer #2: (No Response)

7. PLOS authors have the option to publish the peer review history of their article (what does this mean?). If published, this will include your full peer review and any attached files.

Reviewer #1: **Yes: **Henock Yebyo, University of Zurich, Switzerland

Reviewer #2: No

---

## [Editor Report · Acceptance letter]

17 Jul 2020

PONE-D-20-08295R1 

Dietary diversity and physical activity as risk factors of abdominal obesity among adults in Dilla town, Ethiopia 

Dear Dr. Zeleke:

I'm pleased to inform you that your manuscript has been deemed suitable for publication in PLOS ONE. Congratulations! Your manuscript is now with our production department. 

Kind regards, 

on behalf of

Dr. Sabine Rohrmann 

Academic Editor

PLOS ONE